# Molecular Imaging of Oxygenation Changes during Immunotherapy in Combination with Paclitaxel in Triple Negative Breast Cancer

**DOI:** 10.3390/biomedicines11010125

**Published:** 2023-01-04

**Authors:** Tiara S. Napier, Shannon E. Lynch, Yun Lu, Patrick N. Song, Andrew C. Burns, Anna G. Sorace

**Affiliations:** 1Graduate Biomedical Sciences, University of Alabama at Birmingham, Birmingham, AL 35294, USA; 2Department of Radiology, University of Alabama at Birmingham, Birmingham, AL 35294, USA; 3Department of Biomedical Engineering, University of Alabama at Birmingham, Birmingham, AL 35294, USA; 4O’Neal Comprehensive Cancer Center, University of Alabama at Birmingham, Birmingham, AL 35294, USA

**Keywords:** PET, [^18^F]-FMISO-PET, tumor microenvironment, immunotherapy, anti-PD-1, anti-CTLA-4, 4T1, E0771

## Abstract

Hypoxia is a common feature of the tumor microenvironment, including that of triple-negative breast cancer (TNBC), an aggressive breast cancer subtype with a high five-year mortality rate. Using [^18^F]-fluoromisonidazole (FMISO) positron emission tomography (PET) imaging, we aimed to monitor changes in response to immunotherapy (IMT) with chemotherapy in TNBC. TNBC-tumor-bearing mice received paclitaxel (PTX) ± immune checkpoint inhibitors anti-programmed death 1 and anti-cytotoxic T-lymphocyte 4. FMISO-PET imaging was performed on treatment days 0, 6, and 12. Max and mean standard uptake values (SUV_max_ and SUV_mean_, respectively), histological analyses, and flow cytometry results were compared. FMISO-PET imaging revealed differences in tumor biology between treatment groups prior to tumor volume changes. 4T1 responders showed SUV_mean_ 1.6-fold lower (*p* = 0.02) and 1.8-fold lower (*p* = 0.02) than non-responders on days 6 and 12, respectively. E0771 responders showed SUV_mean_ 3.6-fold lower (*p* = 0.001) and 2.7-fold lower (*p* = 0.03) than non-responders on days 6 and 12, respectively. Immunohistochemical analyses revealed IMT plus PTX decreased hypoxia and proliferation and increased vascularity compared to control. Combination IMT/PTX recovered the loss of CD4+ T-cells observed with single-agent therapies. PET imaging can provide timely, longitudinal data on the TNBC tumor microenvironment, specifically intratumoral hypoxia, predicting therapeutic response to IMT plus chemotherapy.

## 1. Introduction

Triple-negative breast cancer (TNBC) is a molecular breast cancer subtype lacking receptors for estrogen, progesterone, and human epidermal growth factor receptor 2 [1]. TNBC accounts for about 20% of all breast cancer cases worldwide and results in poor clinical outcomes [2,3]. Due to its intra- and intertumoral heterogeneity as well as its lack of actionable molecular targets, limited therapeutic options for TNBC are available [4]. Chemotherapy is the predominant systemic therapy for TNBC, but patient response is often short-lived with metastatic patients showing a median overall survival of 12–18 months [1]. Over the last decade, preclinical studies and clinical trials have demonstrated that compared to other breast cancer subtypes, TNBC has higher immunogenicity, higher enrichment of tumor-infiltrating lymphocytes, and higher levels of programmed cell death ligand 1 (PD-L1) [5]. These findings provide strong evidence for the treatment of TNBC with immunotherapy (IMT), particularly with immune checkpoint inhibitors, such as anti-programmed death 1 (anti-PD1) and anti-cytotoxic T-lymphocyte associated antigen 4 (anti-CTLA4), that disallow negative regulation of T-cell function [6]. Additionally, studies show immune checkpoint inhibitors work better as first-line therapies given in combination with other treatments rather than as later stage, single-agent therapies [7]. In 2019, the United States Food and Drug Administration approved the combination of a chemotherapeutic agent, paclitaxel (PTX), with the first immune checkpoint inhibitor, a monoclonal anti-PDL1 antibody, accepted for TNBC treatment [7]. Still, IMT is not effective for all TNBC patients [1,8,9]. Similarly, PTX is known to not be effective consistently across patients. Given this unmet medical need, current research focuses on better understanding cancer and immune system interactions, including the effect of chemotherapy in conjunction with IMT on the tumor microenvironment (TME).

The TNBC TME is a complex and heterogeneous ecosystem comprised of several different and interacting cell types including endothelial cells, stromal fibroblasts, and a range of immune cells [10]. Within this microenvironment, the rapid proliferation of tumor cells increases the demand for oxygen that cannot be met by surrounding blood vessels. This condition leads to the development of hypoxia which induces tumor cell secretion of proangiogenic factors, such as vascular endothelial growth factor and transforming growth factor-β, for neovascularization [11]. Hypoxic induction of aberrant tumor vasculature has been shown to potentiate tumor development via leaky blood vessels unable to transport immune cells and systemic therapeutics efficiently [11,12]. The hypoxic TME is a key modulator in each step of the metastatic process, a critical factor in immune suppression and chemoresistance [13,14]. In silico results reveal TNBC patients with low hypoxic levels and high immune status demonstrate a significantly better overall survival rate compared to those with high hypoxia and low immune status [15]. Hypoxia has also been shown to drive CD8+ T-cell migration and result in T-cell exhaustion via mitochondrial stress [16,17]. While it is known that hypoxia is a negative component of the TME [18,19], understanding the longitudinal alterations in tumors with respect to the immune response is relatively understudied.

The gold standard for detecting and characterizing tumor hypoxia clinically is the evaluation of tissue post-biopsy with immunohistochemistry or polarographic oxygen-sensitive electrodes; however, these methods are invasive for the patient and rely on samples of tissue volume (~50–100 cells) [20,21]. Moreover, although many prognostic biomarkers have been developed to quantify the immunological characteristics of TNBC, few incorporate hypoxic levels into the stratification of TNBC patients and prediction of response to treatment [15,20,21]. Recent preclinical imaging studies have examined how single-agent, chemotherapy only, or IMT-only treatments alter hypoxic response [21,22]; however, it is unclear how PTX alters the TME when combined with IMT. The importance of elucidating this interaction is underscored by findings that PTX has anti-angiogenic effects and induces vascular remodeling [22,23]. Understanding how hypoxic levels are impacted by combination chemotherapy plus IMT will aid in addressing the challenge of understanding the highly complex TME and translating these findings into relevant, consistent biomarkers for use in the clinical setting.

The present study examines the relationship between hypoxia and tumor response to a combination of chemotherapy and IMT in TNBC. To conduct this investigation, TNBC tumors were established in two syngeneic mouse models, with known differential response kinetics to IMT, 4T1 (IMT-resistant) and E0771 (IMT-sensitive). In vivo PET imaging with [^18^F]-fluoromisonidazole (FMISO) imaging was performed longitudinally to measure hypoxic changes during single-agent and combination therapy. FMISO-PET imaging provides three-dimensional, non-invasive visualization of hypoxia of entire solid tumors [21,24]. Following image acquisition, tumors were extracted at early therapeutic time points to probe for T-cell populations, proliferative status, and vasculature. Quantitative assessment of hypoxic response with FMISO-PET represents a not yet undertaken method to analyze the effect of immune checkpoint blockades plus PTX in TNBC tumors. As combination therapies become increasingly common for cancer treatment, understanding how chemo- and immunotherapies interact with different components of the TME could help guide treatment regimens. Applying advanced molecular imaging techniques to these investigations can also aid in minimizing bias observed in tumor biopsies, which assesses information on a small sample section of the tumor at a single time point. FMISO-PET is an advanced, but clinically available, imaging approach with the potential to identify predictive biomarkers of treatment response to combination chemotherapy and IMT and may be used to understand biological changes during therapies and aid in assessing treatment efficacy for individual TNBC tumors.

## 2. Materials and Methods

### 2.1. Cell Culture

Triple-negative murine mammary carcinoma 4T1 cells were purchased from American Type Culture Collection (Manassas, VA, USA) and cultured in 10% fetal bovine serum in Roswell Park Memorial Institute 1640 complete growth media (Gibco, Waltham, MA, USA). Triple-negative murine mammary carcinoma E0771 cells were purchased from CH3 Biosystems (Buffalo, NY, USA) and cultured in 10% fetal bovine serum in Dulbecco’s Modified Eagle Medium (Gibco) with 2 mM L-glutamine and 1 mM sodium pyruvate. All cells were maintained in a humidified incubator at 37 °C with 5% CO_2_, grown to 80% confluence, and kept at passage numbers <20. Experiments were performed with cells acquired and frozen within 1 month to maintain the phenotype of each cell line.

### 2.2. Syngeneic Mouse Tumor Models

Animal experiments were reviewed and approved by the Institutional Animal Care and Use Committee (IACUC) at the University of Alabama at Birmingham (protocol code 08778, approved 13 March 2020). 2 × 10^5^ 4T1 cells in 100 µL phosphate-buffered saline (PBS) were orthotopically injected into the 3rd mammary fat pad of 5–6 week-old female Balb/c mice (Charles River Laboratories, Wilmington, MA). 5 × 10^5^ E0771 cells in 100 µL PBS were orthotopically injected into the 3rd mammary fat pad of 10–12 week-old female C57Bl6 mice (Jackson Laboratory, Bar Harbor, ME, USA). Primary tumor volume measurements were made with calipers and calculated with the formula: V = (4π/3) × ((shortest diameter × longest diameter × average of shortest and longest diameters)/2). When 4T1 tumors (*n* = 60) reached 80–100 mm^3^ and E0771 tumors (*n* = 37) reached 75–150 mm^3^, mice were enrolled in the study. All mouse procedures and care were completed in accordance with the protocols approved by IACUC.

### 2.3. Treatments

PTX (10 mg/kg; Alfa Aesar, Ward Hill, MA, USA, cat. J62734-MC), anti-PD1 (200 μg), and anti-CTLA4 (100 µg) (Bio X Cell, Lebanon, NH, USA, cat. BE0146 and BE0164, respectively) were administered intraperitoneally with a total injection volume of 100 μL per mouse. Treatments were given as shown in Figure 1.

Thresholds for response to treatment were determined as one standard deviation below tumor volume of control, saline-treated mice. There were 4 cohorts of animals per model and animals were enrolled into long-term treatment response, imaging, or biological validation studies.

### 2.4. [^18^F]-FMISO-PET/CT Imaging and Analysis

[^18^F]-FMISO was generated by the UAB Cyclotron Facility. Tumor-bearing mice (*n* = 32 for 4T1, *n* = 34 for E0771) received 5.55 ± 0.37 MBq [^18^F]-FMISO per mouse in a total injection volume of 100–200 µL sterile saline. The dose was allowed to circulate systemically for 80 min and then 20 min static PET imaging was performed on a GNEXT microPET/CT (SOFIE, Culver City, CA, USA) for mice before treatment on days 0, 6, and 12 as shown in Figure 1. The tumor region of interest (ROI) was selected based on the anatomical CT images. ROIs were reviewed by someone with >10 years’ of experience in preclinical cancer imaging and cancer models. Mean and frequency histogram of standardized uptake value (SUV) were quantified with VivoQuant pre-clinical image processing software (v.4.0; Boston, MA, USA). SUV was calculated as follows:(1)$$\mathrm{SUV}\;=\frac{\begin{array}{c} \mathrm{Measured}\;\mathrm{activity}\;\mathrm{in}\;\mathrm{an}\;\mathrm{ROI}\\ \left(\mathrm{Activity}\;\mathrm{concentration}\;\mathrm{in}\;\mathrm{tissue}\right) \end{array}}{\frac{\mathrm{Injected}\;\mathrm{dose}}{\mathrm{Body}\;\mathrm{weight}}}=\frac{\mathrm{MBq}/\mathrm{mL}}{\mathrm{MBq}/g}=g/\mathrm{mL}.$$

Background ROIs were drawn on the femur muscle of each subject.

### 2.5. Flow Cytometry

Tumor cells were resuspended at a concentration of 10^6^ cells/100 µL and cell surface antigens were stained with fluorophore-conjugated antibodies: NIR-L/D (Invitrogen, Waltham, MA, USA, cat. 17-5321-81), BV510-CD45 (BD, Franklin Lakes, NJ, USA, cat. 563891), PE-Cy7-F4/80 (Invitrogen, cat. 25-4801-82), FITC-CD206 (BioLegend, San Diego, CA, USA, cat. 141704), APC-CD3 (eBioscience, San Diego, CA, USA, cat. 17-0031-82), FITC-CD4 (Thermo Fisher, Waltham, MA, USA, cat. 11-0041-82), PE-Cy5.5-CD8a (ThermoFisher, cat. 45-0081-42), and PE-CD25 (Invitrogen, cat. 12-0251-83). Antibody staining occurred for 30 min at 4 °C in the dark. Cells were acquired on the Attune NxT flow cytometer (Thermo Fisher). Compensation and sequential gating (see Appendix A) were performed with FlowJo software (FlowJo LLC, v.10.8.1; Ashland, OR, USA).

### 2.6. Histological Analysis

Tumor samples were collected and fixed with 10% buffered formalin overnight. Paraffin embedding and the cutting of 5 µm thick tumor slices were performed in the UAB Pathology Core Research Lab. Hematoxylin and eosin (H&E) staining was conducted as previously reported [25]. Immunohistochemical (IHC) analysis was performed to assess hypoxia (pimonidazole; dilution 1:200, Hypoxyprobe, Burlington, MA, USA, cat. HP1-1000), proliferation (Ki67; dilution 1:300, Abcam, Cambridge, MA, USA, cat. ab16667), and vasculature (CD31; Abcam, cat. ab182981). Primary antibodies were incubated overnight at 4 °C. Negative controls were conducted by the omission of primary antibodies. Quantitative analysis of positive IHC staining of the whole section was completed using QuPath v.0.3.0. An artificial neural network (ANN) for pixel classification was used to process images based on color thresholds and probability of positive or negative stain. An automated threshold was used to identify positive and negative regions within tumor slices. Percent of positive staining was determined for each image. The number of vessels and circularity on CD31 fluorescent images were quantified (Fiji ImageJ version 1.53t; National Institutes of Health, Bethesda, MD, USA), and count × circularity was used to generate a vascularity score for quantifying CD31 fluorescence.

### 2.7. Statistics

Unpaired *t*-tests were used to compare treatment groups and non-responder and responder groups. A *p*-value of <0.05 was considered statistically significant. The following notations were used throughout: non-significant, *p* > 0.05; * *p* < 0.05; ** *p* < 0.01; *** *p* < 0.001; **** *p* < 0.0001.

## 3. Results

### 3.1. Combination IMT Plus PTX Lowers Intratumoral Hypoxia as Measured by FMISO-PET Longitudinal Imaging

FMISO-PET showed lower hypoxia in IMT-treated TNBC tumors compared to control as early as day 6 (Figure 2). The SUV_max_ of 4T1 tumors on day 6 was significantly lower in the IMT-only group (1.01) and the IMT plus PTX group (0.99) compared to control (1.45; *p* = 0.04; Figure 2A, B). No significant differences in the SUV_max_ of 4T1-tumor-bearing mice were observed on day 12. Analysis of the top quartile of the histogram of all tumor voxels revealed significantly lower radiotracer uptake, indicative of lower hypoxic level, in the IMT-only (0.25; *p* = 0.009) and IMT/PTX groups (0.26; *p* = 0.01) on day 6 compared to control (Figure 2C). A similar trend was observed on day 12 with IMT-only (0.26; *p* = 0.03) and IMT/PTX-treated groups (0.26; *p* = 0.04) showing significantly lower hypoxic fraction compared to control (0.48).

The SUV_max_ of E0771 tumors on day 12 was significantly lower in the IMT-only (1.04; *p* < 0.0001) and the IMT plus PTX group (1.93; *p* = 0.002) compared to control (3.85; Figure 2D,E). Treatment with the combination IMT/PTX also resulted in lower SUV_max_ compared to treatment with PTX alone (3.34; *p* = 0.02). Analysis of hypoxic fraction in E0771 tumors revealed a significant difference between combination (0.08) and control groups (0.49) on day 6 (*p* = 0.02) and day 12 (0.12 vs. 0.57; *p* = 0.02; Figure 2F). Similar to results seen in analysis of SUV_max_, the combination IMT/PTX resulted in significantly lower intratumoral hypoxia, indicated by hypoxic fraction, compared with PTX (0.45; *p* < 0.05). These changes in hypoxia occur prior to group changes in tumor volume (Figure 3).

### 3.2. Combination IMT Plus PTX Decreases Hypoxia and Proliferation and Increases Vascularity in TNBC Tumors

TNBC tumors were analyzed for biological validation to assess necrosis (H&E), hypoxia (percent of positive pimonidazole), proliferation (percent of positive Ki67), and vascularity (CD31) on days 6 and 12 post-treatment (Figure 3). H&E staining showed less viable tumor tissue in combination IMT/PTX 4T1 tumors compared to control, and these results were confirmed with quantitative analysis (Figure 3A,B). On day 6, PTX-only (52.2%), IMT-only (51.6%), and combination-treated (47.2%) groups all showed significantly lower percent viability compared to control (81.5%; *p* < 0.0001). Similarly, on day 12, PTX-only (52.8%), IMT-only (46.7%), and combination-treated (42.8%) groups all showed significantly lower percent viability compared to control (65.9%; *p* < 0.05). Combination IMT/PTX also decreased viability in E0771 tumors compared to control (Figure 3C,D). Treatment of E0771 tumors with PTX alone (47.1%) and in combination with IMT (36.4%) showed less viability on day 6 compared to control (78.2%; *p* < 0.05). Combination treatment resulted in less hypoxia, indicated by pimonidazole staining, in 4T1 (Figure 3E,F) and E0771 (Figure 3G,H) tumors on days 6 and 12. In 4T1 tumors, IMT alone (4.6%) and with PTX (4.1%) resulted in significantly less hypoxia compared to control on day 6 (29.0%; *p* ≤ 0.0004). Combination treatment also resulted in significantly lower hypoxia than treatment with PTX alone (24.5%; *p* = 0.002). PTX-only (7.8%), IMT-only (5.8%), and combination (6.7%) treatments resulted in significantly lower hypoxia compared to control on day 12 (20.3%; *p* < 0.05; Figure 3F). On day 6, combination-treated E0771 tumors (6.1%) had significantly lower hypoxia compared to PTX-only (16.5%) treated and control tumors (16.6%; *p* < 0.05; Figure 3H). On day 12, combination IMT/PTX resulted in significantly lower hypoxia (0.7%) compared to control (13.9%), PTX alone (13.2%), and IMT-only treated tumors (11.4%; *p* < 0.05).

Staining for Ki67, a marker of proliferation, revealed 4T1 tumors treated with combination IMT/PTX (21.7%) had significantly lower proliferation compared to control on day 12 (42.8%; *p* = 0.02; Figure 3J). Treatment with IMT alone (12.0%) and in combination with PTX (2.9%) resulted in a significantly lower proliferation of E0771 tumors compared to control on day 6 (35.0%; *p* ≤ 0.04; Figure 3L). In 4T1 tumors, IMT alone (15.7) resulted in significantly lower vascularity, as indicated by CD31 staining analyzed for vessel count times circularity, compared to control (22.8; *p* = 0.02; Figure 3N). By day 12, combination-treated 4T1 tumors showed higher vascularity scores (24.4) than control (16.6; *p* = 0.01) and PTX-only tumors (13.6; *p* = 0.0006). On day 6, E0771 tumors treated with combination IMT/PTX showed significantly higher vascularity score (14.1) compared to control (7.4%; *p* = 0.009) and PTX alone treated tumors (9.0; *p* < 0.05; Figure 3P).

### 3.3. PTX Recovers Loss of CD4+ T-Cell Population

Since E0771 tumors have previously demonstrated a higher response to IMT, flow cytometry was performed on day 12 of E0771 tumors to determine alterations in the composition of immune subpopulations resulting from treatment (Figure 4). No significant differences in CD8+ T-cell count were observed between treatment groups; however, the numbers of CD4+ T-cells in the PTX alone (1258; *p* = 0.02) and IMT-only group (1179; *p* = 0.02) were significantly lower than the control group (3557) (Figure 4A,B). CD4+ T-cell count was significantly higher in the combination IMT/PTX-treated group (3111) than in IMT only group (*p* < 0.05; Figure 4A). Compared to control (48.6%), PTX alone (25.7%; *p* < 0.0001) and IMT-only groups (35.2%; *p* = 0.007) showed significantly lower percentages of CD4+ T-cells out of total T-cells (Figure 4C). Combination IMT/PTX-treated tumors (47.2%) had significantly higher percentages of CD4+ T-cells compared to PTX alone (*p* < 0.0001) and IMT-only (*p* = 0.01) treated tumors. No significant differences were observed in M1-phenotype macrophages; however, combination IMT/PTX treatment decreased the number of M2-phenotype macrophages (380 vs. 348; *p* = 0.03; Figure 4D,E). Compared to control, combination IMT/PTX treatment increased the percentage of M1-phenotype macrophages (89 vs. 98%; *p* = 0.03) but decreased the percentage of M2-phenotype macrophages out of total macrophages (10.6 vs. 5.5%; *p* = 0.03; Figure 4F).

### 3.4. FMISO-PET Shows Predictive Value for Response to IMT Plus PTX

To understand the value of FMISO-PET for the prediction of therapeutic response, TNBC tumors were classified as responders or non-responders using end-of-treatment tumor volume (day 18) (day 0; Figure 5). Thresholds for response to treatment were determined as one standard deviation below tumor volume in the control, saline-treated mice, in each of the models. The earliest differences in tumor volume between responders and non-responders were observed on day 15 in the groups receiving IMT either alone or in combination with PTX. In 4T1-tumor-bearing mice (Figure 5A); 3/7 PTX-only treated tumors (*n* = 7) responded to treatment (Figure 5B). Half of the 4T1-tumor-bearing mice (10 out of 20), receiving IMT either alone or in combination with PTX showed tumor response to treatment (Figure 5C,D). A similar trend was seen in the PTX-only treated E0771 tumors with day 15 showing clear separation between responders and non-responders (Figure 5E). Treatment with IMT, either alone or in combination with PTX, resulted in significantly lower tumor volumes in responders compared to non-responders by day 15 (*p* < 0.05; Figure 5F,G). This variation in response is similar to clinical findings of some patients responding effectively and others showing no therapeutic benefit from immunotherapy.

As SUV_mean_ revealed significant differences in treatment groups on days 6 and 12 for both 4T1- and E0771-tumor-bearing mice, differences in SUV_mean_ between responders and non-responders were analyzed (Figure 6). 4T1-tumor-bearing responders showed significantly less intratumoral hypoxia compared to non-responders on day 6 (0.26 vs. 0.42; *p* = 0.02) and day 12 (0.23 vs. 0.41; *p* = 0.02; Figure 6A,B). Similarly, E0771-tumor-bearing responders showed significantly less intratumoral hypoxia compared to non-responders on day 6 (0.14 vs. 0.51; *p* = 0.001) and day 12 (0.14 vs. 0.38; *p* = 0.03; Figure 6C,D). Importantly, these changes in hypoxia were not positively correlated with tumor volumes on these days, revealing a potential mechanism of response to chemotherapy and checkpoint blockade.

## 4. Discussion

In this study, we investigate how chemotherapy alters intratumoral oxygenation when combined with IMT. We demonstrate FMISO-PET imaging provides early prediction of TNBC tumor response to immune checkpoint blockade in conjunction with chemotherapy, revealing a decreased hypoxic tumor fraction in tumors that respond more effectively to treatment (as measured by downstream changes in tumor size). We also examine the direct effects of chemotherapy on necrosis, hypoxia, proliferation, and vascularity, when combined with IMT which are all clinically relevant parameters of TNBC tumors. We demonstrate that compared to single-agent treatment, combination treatment with chemotherapy and IMT recovers the percentage of CD4+ T-cells back to control levels and that combination treatment decreases macrophage polarization to M2-phenotype compared to control. These collective findings are significant as they contribute to understanding how the TME is influenced by a combination of IMT and chemotherapy.

Our results are in congruence with those of Reeves et al. who used FMISO-PET to identify hypoxia in tumor models known to demonstrate a response to IMT. The authors found responding to colorectal and breast tumors showed decreased SUVs compared to non-responders [21]. Zhao et al. also examined the utility of PET imaging with [^18^F]-FMISO for early detection of treatment response to cytotoxic chemotherapies [26]. Using this imaging approach, they monitored hypoxic conditions in MDA-MB-435S breast tumor xenografts in a nude mouse model treated with eribulin, which, like PTX, is a chemotherapeutic agent known to inhibit microtubule formation, which blocks mitosis-causing cell cycle arrest, and ultimately, apoptosis [27,28]. Eribulin-treated tumors showed low hypoxia indicated by FMISO-PET imaging, and this result was validated with a percent of pimonidazole positivity [21]. Further, eribulin increased the amount of microvessels compared to untreated tumors. Aligning with these results were our PET imaging findings, which showed significantly lower hypoxia in 4T1 breast syngeneic tumors treated with chemotherapy alone, and our IHC analyses for pimonidazole and CD31, an endothelial cell marker. Interestingly, in 4T1 tumors, vascularity increased from days 6 to 12 for the two IMT-treated groups and decreased from day 6 to day 12 for the control and PTX-alone-treated groups. Combination-treated E0771 tumors showed the highest measure of vascularity on day 6 compared to the other groups indicating that PTX enhances vascular perfusion. This result aligns with those of Barnes et al. who found PTX increased vascular stability and maturity in mice bearing TNBC tumors established with MDA-MB-231 cells [29]. We note the dosing amount of paclitaxel or other chemotherapies may play a role in whether vasculature is increased or decreased following chemotherapy.

Combination IMT/PTX treatment significantly decreased tumor proliferation, as measured with Ki67 immunohistochemistry early in the course of therapy intervention. This finding suggests the anti-proliferative effect of PTX on tumor cells is heightened when added to IMT. Results also suggest that for 4T1 tumors, and in immune-cold environments, combination treatment takes longer to inhibit proliferation at a rate that is stronger than single-agent chemotherapy. Still, PTX alone treated 4T1 tumors showed the lowest average proliferation as indicated from IHC on day 6. This result aligns with several other published findings demonstrating the anti-proliferative effects of PTX on 4T1 cells [29,30,31]. Quantitative analysis of H&E stained TNBC tumors revealed treatment with combination IMT plus PTX resulted in the highest percentage of necrosis. This result was seen in both tumor lines and on days 6 and 12 and shows that tumors were decreasing in size.

It was hypothesized that significant differences in the CD8+ T-cell counts would be seen between treatment groups based on flow cytometry results; however, none were observed. Instead, differences in CD4+ T-cells were found. Specifically, the effects of PTX or IMT alone versus combination treatment rescued CD4+ T-cells back to levels shown in controls. Combination treatment also decreased M2-like macrophage count, which reveals mechanisms of enhanced biological synergy between treatments as M2-like macrophages have been shown to promote tumor progression and chemotherapeutic resistance [32]. Lastly, we noted that while Reeves et al. found increased interferon gamma (IFN-γ) expression in responders compared to non-responders [21], other studies in E0771 tumors previously found that both IFN-γ and interleukin (IL)-2 expression were heightened by combination IMT/PTX treatment [33]. IFN-γ is associated with anti-proliferative and pro-apoptotic mechanisms and was expected to increase in expression due to chemotherapy. IL-2 is a marker of T-cell activation, expected to be heighted by immune checkpoint blockade. Taken together, these results align with our flow cytometry analyses indicating that the combination IMT/PTX utilizes the immunogenic cell death pathway [33].

The present study is limited by its use of a simplified dosing strategy for the treatment of mice with the combination of IMT plus chemotherapy. Studies have shown that specific dose ranges for immune checkpoint inhibitors used in conjunction with chemotherapy are beneficial; however, varying dosages were not investigated here. Additionally, divergent tumor biology may contribute to the observed differences between tumor response in the two TNBC models utilized for our investigation. These models differed on genetically programmed biases toward Th1 and Th2 response. Future work includes the identification of timing, dosing, and dose sequencing strategies for anti-cancer therapies and parsing response into partial and complete classifications. Future work utilizing FMISO-PET imaging also includes IHC staining for other prominent markers of hypoxia such as hypoxia-inducible factor 1-alpha [34,35]. The use of additional models for TNBC will help increase understanding of the relationship between hypoxia and immune components of the TME. Despite its current limitations, this study demonstrates the utility of FMISO-PET for gaining insight into TME alterations that result from the combination IMT/PTX.

## 5. Conclusions

There are no clinically approved targeted therapies for TNBC. In addition to the lack of targeted therapies, there are no predictive markers that robustly quantify who will benefit from treatment with IMT which inhibits effective personalization and optimization of patient treatment regimens. The present study examines a clinically relevant approach to better understand underlying biological changes in the tumor during combination therapy that could help personalize and optimize novel combination IMT/chemotherapy for TNBC. Results from this study show that noninvasive FMISO-PET imaging yields quantitative data on hypoxia in individual TNBC tumors that can be used to predict tumor response to combination therapy. Specifically, PTX enhances immune response creating a more susceptible environment for the IMT mechanism of action. These findings aid in understanding biologically distinct features of the TME and personalizing anti-cancer therapies.

## Figures and Tables

**Figure 1 biomedicines-11-00125-f001:**
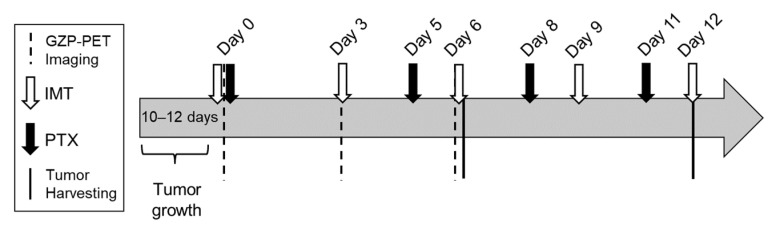
Experimental timeline showing tumor implantation, imaging, and treatment schedule.

**Figure 2 biomedicines-11-00125-f002:**
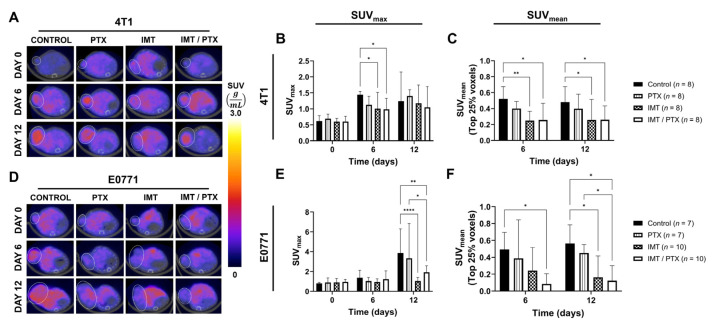
FMISO-PET imaging provides quantitative data on TNBC tumor response. (**A**) Representative FMISO-PET images of 4T1-tumor-bearing mice on days 6 and 12 show differences in intratumoral hypoxia between treatment groups. (**B**) SUV_max_ was significantly lower in IMT ± PTX treated 4T1 tumors on day 6 compared to control, indicating treatment with IMT results in lower hypoxic levels in the tumor ROI. (**C**) SUV_mean_ was significantly lower in IMT ± PTX treated 4T1 tumors on days 6 and 12 compared to control, indicating IMT treatment results in low hypoxic levels in the tumor ROI. (**D**) Representative FMISO-PET images of E0771-tumor-bearing mice on days 6 and 12 show differences in intratumoral hypoxia between treatment groups. (**E**) SUV_max_ was significantly lower in IMT± PTX treated E0771 tumors on day 12 compared to control, indicating treatment with IMT results in lower hypoxic levels in the tumor ROI. (**F**) SUV_mean_ was significantly lower in IMT± PTX treated E0771 tumors on days 6 and 12 compared to control, indicating IMT treatment results in low hypoxic levels in the tumor ROI. Mean ± SD (* indicates *p* < 0.05, ** indicates *p* < 0.01; **** indicates *p* < 0.0001; multiple *t*-tests).

**Figure 3 biomedicines-11-00125-f003:**
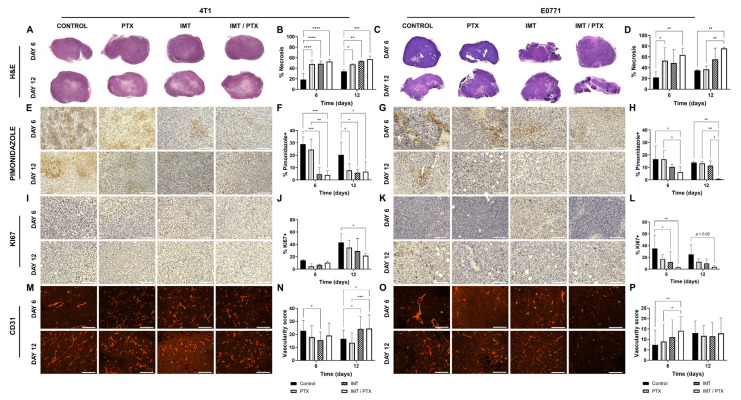
Histochemical imaging and quantitative analyses performed on TNBC tumors validate findings from analysis of FMISO-PET imaging data. (**A**–**D**) 4T1 and E0771 tumors were harvested on days 6 and 12, sliced, and stained with H&E and percentage of viable tumor tissue was quantified. Tumors were also stained and quantified for (**E**–**H**) pimonidazole, a marker for hypoxia, (**I**–**L**) Ki67, a marker for proliferation, and (**M**–**P**) CD31, a marker for vasculature (20× magnification; scale: 125 µm). Mean ± SD (* indicates *p* < 0.05; ** indicates *p* < 0.01; *** indicates *p* < 0.001; **** indicates *p* < 0.0001; multiple *t*-tests).

**Figure 4 biomedicines-11-00125-f004:**
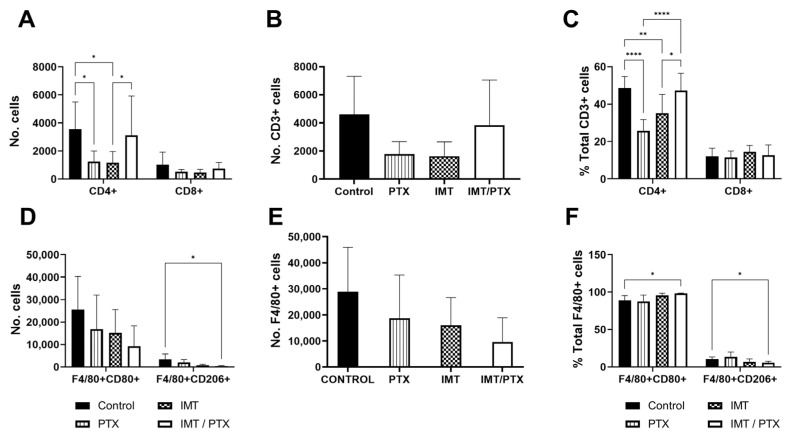
PTX is shown to recover loss of CD4+ T cell population and combination IMT/PTX significantly decreases the number of M2-phenotype macrophages. T-cell and macrophage panels were utilized with flow cytometry performed on E0771 tumors extracted on day 12 (*n* = 4/group) to distinguish subpopulations of (**A**–**C**) T-cells and (**D**–**F**) macrophages. Additional data shown in Appendix A. Mean ± SD (* indicates *p* < 0.05; ** indicates *p* < 0.01; **** indicates *p* < 0.0001; multiple *t*-tests).

**Figure 5 biomedicines-11-00125-f005:**
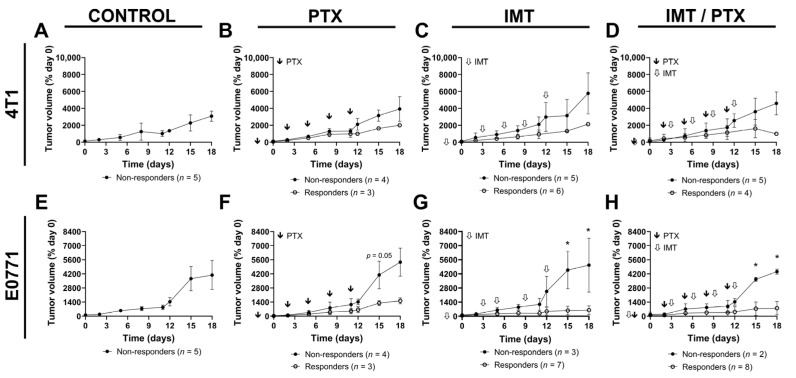
Mice were sorted into non-responder and responder groups based on tumor volume. Mice bearing (**A**–**D**) 4T1 and (**E**–**H**) E0771 tumors were separated into non-responder and responder groups for the purpose of analyzing FMISO-PET predictive value. Mean ± SD (* indicates *p* < 0.05; multiple *t*-tests).

**Figure 6 biomedicines-11-00125-f006:**
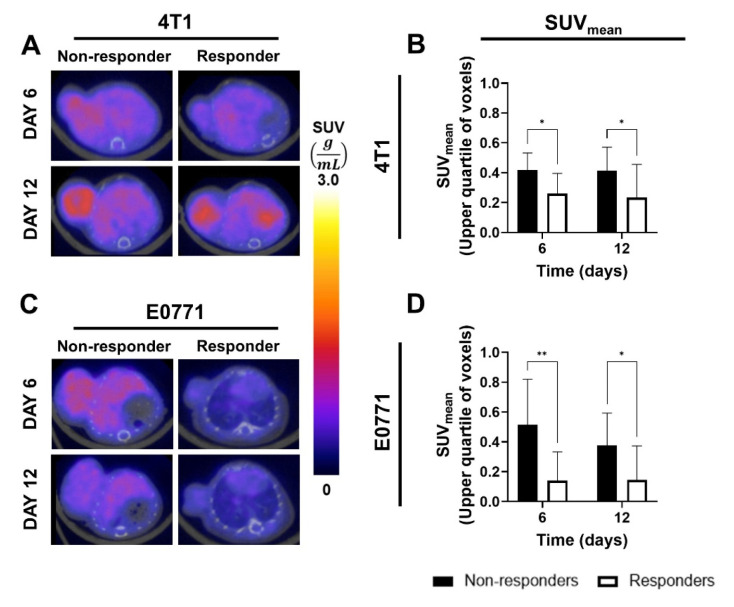
FMISO-PET imaging of hypoxia accurately predicts response to IMT and chemotherapy in TNBC tumors. Representative FMISO-PET images are shown of a nonresponder and responder (as measured by end tumor volume) within the combination IMT/PTX-treated mice and SUV_mean_ of the upper quartile of voxels from (**A**) 4T1 and (**C**) E0771 tumors. SUV_mean_ of the upper quartile of voxels from (**B**) 4T1 and (**D**) E0771 tumors reveal reduced hypoxia in tumors that respond better to treatment prior to changes in tumor size, seen on days 18 (4T1) and 15 (E0771). Mean ± SD (* indicates *p* < 0.05, ** indicates *p* < 0.01; multiple *t*-tests).

## Data Availability

Datasets and materials are available upon reasonable request to the corresponding author.

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
