# Peer review of "Molecular Imaging of Oxygenation Changes during Immunotherapy in Combination with Paclitaxel in Triple Negative Breast Cancer"

_biomedicines, 2023, doi:10.3390/biomedicines11010125_

Round 1

Reviewer 1 Report

The authors Napier et al investigated the Molecular Imaging of Oxygenation Changes during Immunotherapy in combination with Paclitaxel in Triple Negative Breast Cancer. Using [18F]-fluoromisonidazole (FMISO) positron emission tomography (PET) imaging, they monitored the changes in response to immunotherapy (IMT) with chemotherapy in TNBC. They showed IMT plus PTX combination decreased hypoxia and proliferation, increased vascularity and recovered the loss of CD4+ T-cells. The studies presented are interesting and well-written.

Major comments

1) The authors majorly focused on FMISO-PET imaging. As HIF-1α, BNIP3, PDK1, and GLUT1 are the prominent hypoxia markers, I suggest authors analyze the protein expression of any of these markers by IHC staining, which would support the PET imaging results.

2) I would suggest authors present all the flow cytometry data and mention the complete gating strategy in supplementary and say in the figure legend.

3) The Figure 3 resolution is poor; it is hard to visualize the changes in H&E and IHC images. I suggest authors improve the resolution of the images.

Reviewer 2 Report

Review of the manuscript entitled:

Molecular Imaging of Oxygenation Changes during Immunotherapy in Combination with Paclitaxel in Triple Negative Breast Cancer

submitted to MDPI Biomedicines biomedicines-2073176

Summary:

Authors in their current manuscript examined the biological changes in the tumor during combination therapy that could help personalize and optimize novel combination IMT/chemotherapy for Triple Negative Breast Cancer. They showed that noninvasive FMISO-PET imaging yields quantitative data on hypoxia in individual TNBC tumors that can be used to predict tumor response to combination therapy. They confirmed that PTX enhances immune response creating a more susceptible environment for IMT mechanism of action.

Strengths of the manuscript:

The selection of tumors is also appropriate, as breast cancer and breast adenocarcinoma cases are steadily increasing nowadays and have high mortality rates. I would like to highlight the diversity of the methods used. In vivo, ex vivo and histochemical imaging studies have been extremely well developed and performed. The manuscript has a logical structure, the description of the methods are detailed and reproducible. The conclusion is correct. The manuscript contains all the necessary additional information.  Illustrations, figures, schemes are in very good quality.

Due to the potential of the topic and its elaboration, I recommend it for publication as is.

Author Response

Thank you for your comments and positive feedback. 

Round 2

Reviewer 1 Report

The manuscript is considered suitable for the publication.